# Effects of music intervention combined with progressive muscle relaxation on anxiety, depression, stress and quality of life among women with cancer receiving chemotherapy: A pilot randomized controlled trial

Khanh Thi Nguyen[1,2], Huong T. X. Hoang[3], Quang V. Bui[4], Dorothy N. S. Chan[1], Kai C. Choi[1], Carmen W. H. Chan[1] *

1 The Nethersole School of Nursing, The Chinese University of Hong Kong, Hong Kong SAR, China, 2 Nam Dinh University of Nursing, Nam Dinh, Vietnam, 3 Phenikaa University, Hanoi, Vietnam, 4 Hanoi Oncology Hospital, Hanoi, Vietnam

* whchan@cuhk.edu.hk

## Abstract

Beneficial effects of music intervention and progressive muscle relaxation alone on psychological issues were reported, however, studies evaluating their combined effects are limited. This study aimed to investigate the feasibility, acceptability and preliminary effects of music intervention combined with progressive muscle relaxation on anxiety, depression, stress, and quality of life among breast and gynaecological cancer patients receiving chemotherapy. **Methods:** The study was carried out from March to May 2022 in an oncology hospital in Vietnam. A single-blinded randomized controlled trial was conducted among 24 women with breast and gynaecological cancer undergoing chemotherapy. The intervention group (n = 12) received a face-to-face training program about music listening and progressive muscle relaxation skills. They then performed the self-practice daily at home for three weeks. The control group (n = 12) received standard care, including health assessment, regular health advice and nutrition consultation. Ten participants in the intervention group were interviewed with open-ended questions to explore the acceptability of the intervention. Anxiety, depression and stress were measured using the Depression Anxiety Stress Scale, while The Functional Assessment of Cancer Therapy–General was used to evaluate the quality of life. The outcome measurements were collected at baseline (T0), post-intervention (3rd week, T1) and follow-up (6th week, T2). Appropriate descriptive statistics were used to depict the outcome measures across study time points. **Results:** A total of 24 patients were eligible to join, and 20 of them completed the study. Greater reductions in anxiety, depression and stress were observed in the intervention group than in the control group at T1 and T2. Greater improvements on quality of life were found in the intervention group than control group at T1 and T2 with respect to T0. The content analysis supported the acceptable intervention of participants through two themes, perceived beneficial effects on psychological and physical health and willingness to keep practising in the future. **Conclusions:** Implementing music intervention combined with progressive muscle relaxation is feasible and

**Data Availability Statement:** The data underlying the results presented in the study are available at https://doi.org/10.7910/DVN/6I62QK.

**Funding:** The author(s) received no specific funding for this work.

**Competing interests:** The authors have declared that no competing interests exist.

had a trend in reducing anxiety, depression and stress levels. A larger scale randomized controlled trial is needed to confirm the effect of the intervention on outcomes.

**Trial registration:** This trial was registered on ClinicalTrials.gov with ID: NCT05262621.

## 1. Introduction

Cancer patients receiving chemotherapeutic treatments routinely suffer from distressing physical symptoms such as gastrointestinal problems, fatigue, neurological effects [1] and psychological symptoms, including anxiety and depression, which decrease their quality of life (QoL) [2]. It was reported that 26.9% and 41.55% of breast cancer patients taking chemotherapy experienced anxiety and depression, respectively [3]. A significant prevalence of cases of anxiety (38%) and depression (33%) was found in ovarian cancer patients at the end of chemotherapy [4]. Thus, psychological support for breast and gynaecological cancer (BGC) patients is essential in nursing care.

Recently, non-pharmacological interventions have been commonly suggested to minimize the adverse psychological effects during cancer treatment. Many non-pharmacological interventions were suggested to reduce chemotherapy-related stress, such as cognitive behavioural therapy, problem-solving, yoga [5], distraction, music intervention (MI), virtual reality [6], and progressive muscle relaxation (PMR) [7–9]. Among these interventions, MI and PMR are more feasible to implement than others because they are inexpensive, safe and acceptable [9–12].

MI is when music, as an agent of change, is used to establish a therapeutic relationship, to help individuals reach goals related to physical, emotional, cognitive, and social needs [13]. Systematic reviews indicated that MI was an effective method to relieve anxiety and improve cancer patients' QoL during chemotherapy [14,15]. Especially, anxiety and depression reduction were observed among breast cancer patients taking chemotherapy after receiving MI [11,14,16]. PMR is a deep relaxation technique which involves the individual alternately tightening and relaxing various muscle groups in the body. PMR is commonly used for stress management, significantly reducing psychological stress, including anxiety, in breast cancer patients receiving chemotherapy [8,9].

Many studies evaluated the effect of MI or PMR as a single intervention, but limited studies evaluated their combined effects on patients' psychological well-being. Previous studies confirmed that MI combined with PMR had a more significant impact on emotional well-being than MI, and PMR applied alone [17–19]. Two studies compared MI and PMR alone with MI combined with PMR among university students, reporting that the integrated intervention had a better reduction in anxiety, relaxation, and stress [17,18]. However, these studies assessed only a 15-minute-single session intervention, and outcomes were evaluated immediately post-interventions without assessing long-term effects. Moreover, music was used as music background to assist PMR, but it was not worked independently. Zhou, Li [20] reported significant effects of MI combined with PMR on anxiety and depression among breast cancer patients after radical mastectomy. The study tested the combined intervention to manage mental disorders during hospitalization, while it is evident that BGC patients had higher anxiety and depression scores during chemotherapy, and the issues continued over time, even during surviving periods [21]. Thus, the strategies by which BGC patients can self-manage their psychological issues after discharge should be considered. Liao, Wu [19] confirmed that 8-week home-based MI plus PMR had a more significant effect on depression than PMR alone among cancer patients. But no control group was designed in Liao's study to identify the effect

of the combined intervention on the outcomes. To narrow the above research gap, we conducted the current pilot study to:

- Test the feasibility and acceptability of MI combined with PMR in BGC patients receiving chemotherapy.

- Evaluate the preliminary effects of this combined intervention on anxiety, depression, stress, and QoL among BGC patients receiving chemotherapy.

This is the first study testing the effects of MI combined with PMR in the short and long term on psychological outcomes in patients' home settings, particularly focusing on BGC patients receiving chemotherapy.

## 2. Methods

### 2.1. Design

A 2-arm, assessor-blinded pilot randomized controlled trial was conducted at at a chemotherapy unit in an oncology hospital, Hanoi, Vietnam, from March to May 2022.

### 2.2. Participants

The inclusion criteria were: (1) women with breast or gynaecological cancer aged 18 years or older; (2) receiving chemotherapy once every three weeks and have at least three chemotherapy cycles left; (3) Have Karnofsky score $\geq$ 80 [22]; (4) able to communicate, read and write in Vietnamese; (5) have a device such as a smartphone, MP3 to keep the audio file; (6) consent to join the study. We excluded: (1) the patients who cannot understand the study procedures; (2) those with mental health illnesses, deafness, or blindness; (3) receiving other treatments or therapy to manage anxiety or depression, (4) with a potential treatment (e.g. surgery) or personal plans that prevent them from practising daily during the six weeks of the intervention.

### 2.3. Study sample

Convenience sampling was used to recruit the participants. The sample size planning followed a rule of thumb for determining an appropriate sample size in which the outcome is a continuous measurement. Julious [23] recommended that for the evaluation of a feasibility study with 2-arms, a minimum of 12 participants in each arm should be considered.

### 2.4. Randomization and allocation concealment

Blocked randomization procedures with a block size of 8 in 1:1 allocation ratio was performed by an independent researcher who did not involve any steps of the intervention program. A random group allocation list was generated by using the online randomizer (https://www.sealedenvelope.com/simple-randomiser/v1/lists). The group identifiers corresponding to the list were sealed in sequentially numbered, and opaque envelopes. To avoid tampering, the creator of the envelopes signed on the back of the envelopes [24]. The person who prepared and kept the envelopes was blinded and not involved in recruiting participants to prevent ordering participants into groups [25]. Patients receiving chemotherapy and meeting the inclusion criteria were identified by the research assistant, who was an experienced nurse in the chemotherapy unit. The research assistant contacted the participants. Patients who agreed to participate in the study signed the consent form. They were then asked to complete the baseline measurements before randomization.

## 2.5. The combined intervention PMR and MI procedure

The participants in the intervention group were required to wear loose-fitting, soft clothes, and entered a private, quiet, dimmed light room to receive a face-to-face training session of PMR combined with MI. The training session was performed by the principal investigator (PI), who is certified in PMR and MI. After being introduced to the program, the effects of MI and PMR on psychological issues, the patients were trained on PMR skills and listening to music. The PI demonstrated step-by-step instructions on PMR, and then the participants performed the return demonstration. The participants were instructed on tensing and relaxing 16 muscle groups in sequence, following face and neck, arms and hands, chest and abdomen, legs and feet (see S1 File). Each muscle group tightened for five to seven seconds and then suddenly relaxed for 20–30 seconds. Each muscle group was performed twice [26]. After training PMR skills, the intervener asked participants to select their preferred music from a list including folk songs, religious music, revolutionary music and Vietnamese bolero music. These music genres were selected based on the findings of a qualitative study conducted by Nguyen, Vu [27] for the types of music preferred by breast and gynaecological cancer patients. The researcher pre-prepared the list of music tracks with relaxing music with slow, consistent rhythmic, soft melody [14] and a tempo of 60 to 80 beats/min [11,16,28]. Instrumental music was used because it was more effective in reducing anxiety than vocal music [29]. After selecting the preferred music, the PI turned on the MP3 file consisting of 20 minutes of PMR instruction and, right after that, 20 minutes of the patient's selected music. The participants were asked to lie down, close their eyes, listen to the recorded instruction in the MP3 file and self-practice under the supervision of PI. After the training, the PI asked participants about any issues or difficulties during their practice and encouraged them to self-practice once a day when they feel the most comfortable at home. The patients' selected MP3 files were sent to their phone via the Zalo application. To remind the skills for the participants at home, a structured guideline and coloured pictures presenting the steps of the PMR were distributed to them. A "Self-practice Record Form (SRF)" was given to participants to report their adherence, any adverse events, and the reasons for non-adherence. The PI arranged a meeting with a family member to explain the benefits of the intervention and assigned them to remind and encourage the participants to practice every day at home. Once every week, a research assistant called participants to encourage them to keep practising, remind filling SRF daily and ask their feelings about the practice.

The control group received standard care, including health assessment, regular health advice and nutrition consultation during chemotherapy by doctors and nurses. To minimize the bias by contacting, weekly phone calls were also carried out to participants to ask about their general health and remind them to return to the hospital in subsequent chemotherapy on time.

## 2.6. Outcome measures and instruments

The recruitment rate, eligibility, consent rate, refusal rate, reasons for refusal, and attrition rate were recorded to assess the feasibility. Individual face-to-face semi-structured interviews were conducted with ten participants in the intervention group to review the acceptability of the intervention. The interview lasted 20 to 30 minutes which covered four questions:

1. Could you describe your experience after three weeks of practising the intervention?

2. How did you feel after practising?

3. Could you give me some comments or suggestions about the program?

4. Would you want to keep practising the intervention in the future?

Participants' demographic characteristic was collected using a demographic sheet which included age, marital status, education level, employment, family income, stage of cancer, frequency of chemotherapy, and treatment regimen.

Anxiety, depression and stress were measured using the short form of the Depression Anxiety Stress Scale (DASS-21) [30]. The instrument was translated and used in the Vietnamese population with good internal consistency (Cronbach's alpha: depression subscale-0.72; Anxiety subscale-0.77; and Stress subscale-0.70, overall scale-0.88) [31]. The scale includes 21 items, with seven items for Anxiety (DASS21-A), seven items for depression (DASS21-D) and seven items for stress (DASS21-S). Each item was measured on a 4-point Likert-type scale ranging from 0 = "Do not apply to me at all" to 3 = "Applied to me very much or most of the time". Scores on the DASS-21 needed to be multiplied by 2 to calculate the final score. The score of each subscale ranged from 0 to 42. The higher the score is, the more severe is the anxiety, depression and stress level.

The Functional Assessment of Cancer Therapy–General (FACT-G) was a 27-item self-report measure used to assess QoL in cancer patients, with Cronbach's alpha for the scale being 0.89 [32]. This tool has been translated into Vietnamese by the FACIT system. It evaluates on physical, social, emotional, and functional well-being. Each item is scored in five categories from 0 to 4, meaning 'Not at all, 'A little bit', 'some what', 'Quite a bit' 'Very much'. A total FACT-G score was computed by adding subscale scores. The score ranged from 0 to 108; the higher the score, the better the QOL.

## 2.7. Data collection

The baseline data was collected using the demographic sheet, DASS, and FACT-G by an outcome assessor. The outcome assessor was blinded to the study groups to collect data and did not involve any study steps. DASS and FACT-G were re-evaluated after finishing three weeks of self-practice intervention (T1) and three weeks after T1 (T2). The interview to review the acceptability of the intervention was conducted at T1 (see Table 1).

## 2.8. Data analysis

Statistical analyses were performed using SPSS version 25. Appropriate descriptive statistics, such as mean, standard deviation, frequency and percentage, were used to summarize and present the baseline characteristics of the participants and the outcome measures across study time points. The preliminary effects of the intervention on the outcomes were estimated by the bias-corrected Hedges'g effect sizes [33], together with 95% confidence intervals, based on the standardized mean differences of the mean changes of the outcomes at T1 and T2 with respect to T0 between the intervention and control groups.

**Table 1. Time points for data collection.**

|  | Screening | Week 0 Baseline | Week 1–3 | Week 3 | Week 6 |
|---|---|---|---|---|---|
| Number of eligible participants | X |  |  |  |  |
| Number of participants sign the consent form | X |  |  |  |  |
| Reason for ineligibility, refusal | X |  |  |  |  |
| Karnofsky performance status scale | X |  |  |  |  |
| DASS-21 |  | X |  | X | X |
| FACT-G |  | X |  | X | X |
| SRF |  |  | X |  |  |

Every interview was recorded, followed by exact transcription and managed with Nvivo 12 software. The interview transcriptions were analysed using a qualitative content analysis methodology following Graneheim and Lundman [34]. The text was broken down into distinct meaning segments; these were then distilled into concise units of meaning and given a code. The differences and similarities of the codes were examed to be classified into sub-categories, categories, and themes [35].

## 2.9. Ethical approval

The pilot study was conducted in accordance with the Declaration of Helsinki and written informed consent was obtained from all participants prior to enrollment. The study was approved by The Joint Chinese University of Hong Kong—New Territories East Cluster Clinical Research Ethics Committee (No. 2021.725), the Ethics Committee in medical research, Nam Dinh University of Nursing, Vietnam, Vietnamese Ministry of Health (No. 2792).

## 3. Result

### 3.1. Feasibility of the study

The whole data collection process was done from 8th March to 2nd May 2022, in which the total time for recruitment was 14 days. During the recruitment period, 126 patients were assessed for eligibility, of which 94 (74.6%) patients were excluded because of ineligibility. After excluding eight patients with reasons: positive with Covid-19 (one patient), refused (three patients), had Karnofsky score lower than 80 (two patients), and could not arrange time for the intervention (two patients). The remaining 24 consented participants were randomized into the intervention group (n = 12) or the control group (n = 12).

One participant dropped out (4.2%) at T1 because she changed to another hospital. Three participants dropped out at T2 (12.5%), including two participants in the intervention group due to changing the chemotherapy schedule and feeling exhausted and one in the control group because of the changing chemotherapy schedule (see details in Fig 1).

### 3.2. Preliminary effects of MI combined with PMR on outcome measures

**3.2.1 Participants' demographic and clinical characteristics.** The mean age of the participants was 52.3 years. Nearly half of the participants (45.8%) obtained a bachelor's or higher degree. The majority of participants did not have a religious belief (83.3%). More than half of the participants were current workers (58.3%), living in a city area (66.7%), and married (62.5%). The proportion of people with an average income/ person/ month lower than 210 USD accounted for 54.2% (Table 2).

Breast cancer accounted for most of the participants (79.2%). Nearly half of participants were stage 4 cancer (45.8%). The mean time from diagnosis was 20.04 months. Half of the participants were treated with docetaxel, whereas paclitaxel was treated for another 29.2%. Other chemotherapy regimens, such as gemcitabine, vinorelbine, and capecitabine, accounted for 20.8% (see detail in Table 3).

**3.2.2. Preliminary effects of the intervention on outcome measures across time.**
Table 4 shows the means of outcomes at the study time points and mean changes from baseline between the intervention and control groups. On average, the participants in the intervention group reported reductions on anxiety and depression at T1 and T2 relative to T0, while the participants in the control group had an increase in these outcomes. The Hedges' g effect sizes on anxiety and depression were respectively 0.57 (95% CI: -0.20 to 1.35) and 0.21 (95% CI: -0.55 to 0.97) at T1 and 0.52 (95% CI: -0.30 to 1.34) and 0.60 (95% CI: -0.23 to 1.42) at T2.

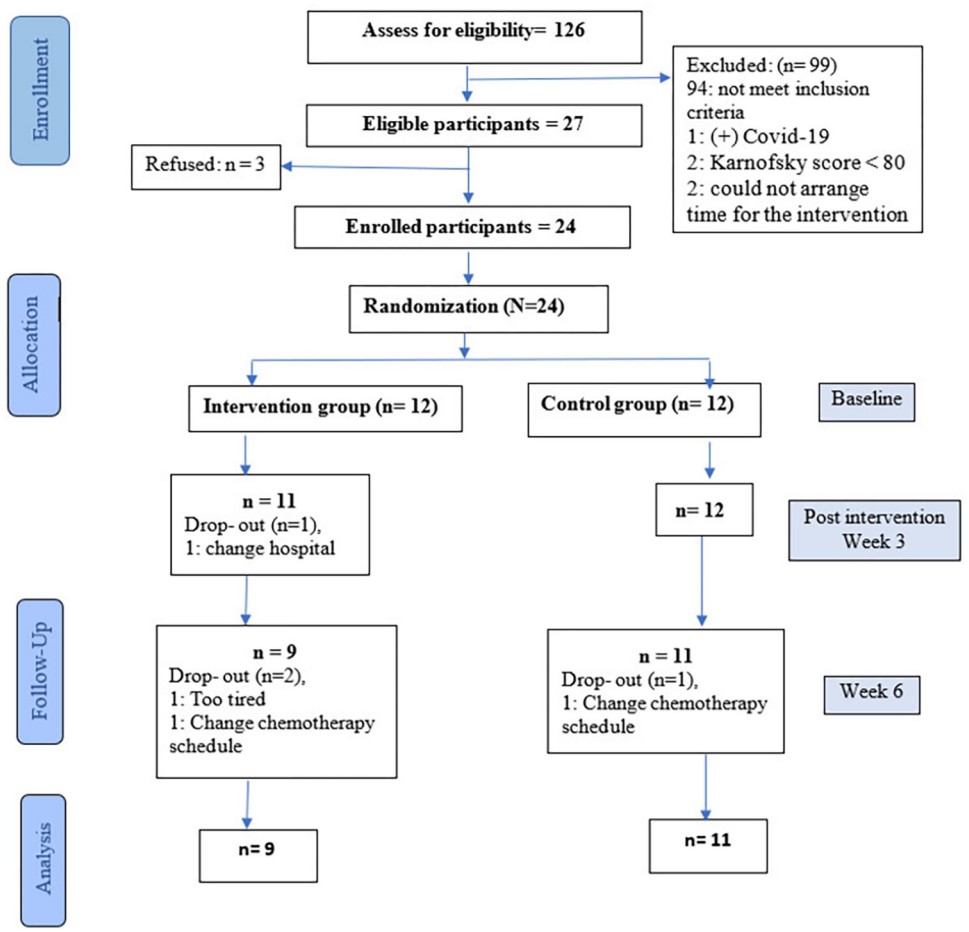

**Fig 1. The Consolidated Standards of Reporting Trial (CONSORT) flow diagram.**

Similar trends were also noted on stress at T1 (g = 0.32, 95% CI: -0.44 to 1.08). However, both the groups reported reduction stress level at T2 relative to T0, with greater reduction observed in the intervention group (g = 0.01, 95% CI: -0.79 to 0.82). For QoL, the intervention group had improvements at both T1 and T2, whereas the control group had decrements (g = 0.43, 95% CI: -0.33 to 1.20 at T1 and g = 0.82, 95% CI: -0.03 to 1.66 at T2).

### 3.3. The acceptability of the intervention

A total of ten participants completed the semi-structured interview, and two themes were generated regarding the acceptability of the intervention. The themes were: (1) Perceived beneficial effect on psychological and physical health and (2) Willing to keep practising in the future.

### Theme 1: Perceived beneficial effect on psychological and physical health

Most participants reported that they had better psychological and physical health after three weeks of self-practice at home. Participants reported feeling comfortable, relaxed, and calm.

"*I feel more comfortable in my body, more relaxed, and my mind is more relaxed (P2)*".

**Table 2. Baseline demographic information of the participants by group assignment.**

| Variables | Total (N = 24) | Group assignment | |
|---|---|---|---|
| | | MCP group (n = 12) | Control group (n = 12) |
| **Age** (years) Mean (SD) [Range] | 52.29 (13.44) [31–75] | 49.00 (11.75) [31–74] | 55.58(14.69) [33–75] |
| **Education** | | | |
| Secondary school and lower | 9 (37.5%) | 4 (33.3%) | 5 (41.7%) |
| High, vocational school | 4 (16.7%) | 1 (8.3%) | 3 (25%) |
| Bachelor or higher | 11 (45.8%) | 7 (58.3%) | 4 (33.3%) |
| **Religion** | | | |
| No religion | 20 (83.3%) | 11 (91.7%) | 9 (75%) |
| Buddism | 3 (12.5%) | 0 (0%) | 3 (25%) |
| Christianity | 1 (4.2%) | 1 (8.3%) | 0 (0%) |
| **Employment status** | | | |
| Currently employed | 14 (58.3%) | 9 (75%) | 5 (41.7%) |
| Currently not employed | 10 (41.7%) | 3 (25%) | 7 (58.3%) |
| **Place of living** | | | |
| City | 16 (66.7%) | 8 (66.7%) | 8 (66.7%) |
| Village | 8 (33.3%) | 4 (33.3%) | 4 (33.3%) |
| **Marital status** | | | |
| Married | 15 (62.5%) | 8 (66.7%) | 7 (58.3%) |
| Single/ Divorced/Separated/ Widowed | 9 (37.5%) | 4 (33.3%) | 5 (41.7%) |
| **Income** | | | |
| >5 million VND ($210) | 13 (54.2%) | 7 (58.3%) | 6 (50%) |
| 5–10 million VND | 9 (33.3%) | 4 (25%) | 5 (41.7%) |
| < 10 million VND | 2 (8.3%) | 1 (8.3%) | 1 (8.3%) |

MCP: Music intervention combined with progressive muscle relaxation group, SD: standard deviation VND: Vietnam Dong

**Table 3. Baseline clinical characteristics of the participants by group assignment.**

| Variables | Total (N = 24) | Group assignment | |
|---|---|---|---|
| | | MCP group (n = 12) | Control group (n = 12) |
| **Cancer Diagnosis** | | | |
| Breast cancer | 19 (79.2%) | 12 (100%) | 7 (58.3%) |
| Gynaecological cancer | 5 (20.8%) | 0 (0%) | 5 (41.7%) |
| **Cancer stage** | | | |
| Stage 1 | 3 (12.5%) | 1(8.3%) | 2 (16.7%) |
| Stage 2 | 6 (25%) | 4 (33.3%) | 2 (16.7%) |
| Stage 3 | 4 (16.7%) | 2 (16.7%) | 2 (16.7%) |
| Stage 4 | 11(45.8%) | 5 (41.7%) | 6 (50%) |
| **Time from diagnosis (Months) Mean (SD)[Range]** | 20.04 (28.0) [1–108] | 10.29 (10.47) [3–36] | 29.17 (36.72) [1–108] |
| **Chemotherapy Regimen** | | | |
| Docetaxel | 12 (50%) | 8 (66.7%) | 4 (33.3%) |
| Paclitaxel | 7 (29.2%) | 3 (25%) | 4 (33.3%) |
| Others | 5 (20.8%) | 1 (8.3%) | 4 (33.3%) |

MCP: Music intervention combined with progressive muscle relaxation group, SD: Standard deviation.

**Table 4. Outcome measures across time between the groups.**

| Outcomtes | Intervention group | Control group | Effect size (95% CI)# |
|---|---|---|---|
| **Anxiety score (range: 0–42)** | | | |
| T0 | 9.67(4.81) | 9.83(11.07) | |
| T1 | 8.73 (9.64) | 13.83 (11.68) | |
| T2 | 8.22 (8.45) | 14.0 (13.02) | |
| Change (T1-T0) | -1.27 (9.93) | 4.0 (6.98) | 0.57 (-0.20, 1.35) |
| Change (T2-T0) | -1.56 (5.36) | 3.45 (10.81) | 0.52 (-0.30, 1.34) |
| **Depression score (range: 0–42)** | | | |
| T0 | 7.17 (8.07) | 10.67 (9.99) | |
| T1 | 7.45 (10.55) | 11.83 (12.52) | |
| T2 | 7.11 (10.68) | 12.9 (12.27) | |
| Change (T1-T0) | -1.18 (12.44) | 1.17 (7.26) | 0.21 (-0.55, 0.97) |
| Change (T2-T0) | -0.67 (3.46) | 2.18 (4.93) | 0.60 (-0.23, 1.42) |
| **Stress score (range: 0–42)** | | | |
| T0 | 11.67 (5.71) | 15.17 (9.08) | |
| T1 | 11.09 (11.36) | 17.33 (12.49) | |
| T2 | 12 (8.06) | 14.9 (13.19) | |
| Change (T1-T0) | -1.27 (12.2) | 2.17 (7.26) | 0.32 (-0.44, 1.08) |
| Change (T2-T0) | -0.67 (6.25) | -0.55 (9.96) | 0.01 (-0.79, 0.82) |
| **QOL overall (range: 0–108)** | | | |
| T0 | 78.51 (12.25) | 76.55 (17.3) | |
| T1 | 77.48 (16.26) | 71.1 (18.85) | |
| T2 | 82.35 (15.46) | 69.28 (23.87) | |
| Change (T1-T0) | 0.38 (13.89) | - 5.44 (10.83) | 0.43 (-0.33, 1.20) |
| Change (T2-T0) | 4.34 (13.6) | -6.41 (10.49) | 0.82 (-0.03, 1.66) |

Data are presented as mean (standard deviation) at time points.

Abbreviations: T0, baseline; T1, 3 weeks from baseline; T2, 6 weeks from baseline, CI: Confidence Interval.

# Small sample bias-corrected Hedges'g effect size which corresponds to the standardized mean difference of the mean changes at the underlying time point with respect to T0 between the intervention and control groups.

Besides, some participants reported positive effects on physical health, such as reduced nausea, vomiting, pain, and insomnia.

*"During the practice, I was nauseous and uncomfortable; at that time, I just focused, and then I got better and forgot the feeling of nausea. After that, I kept focusing on the forehead area, focusing on it all, focusing on only one position. My legs were the same, my arms were the same, my breathing was the same, and I didn't think about things (P3)."*

## Theme 2: Willing to keep practising in the future

The participants could feel the positive impacts on their health after the intervention. All the participants were willing to continue practising the intervention in the future and wanted to share the intervention with other cancer patients.

*"I think I will practice often; for example, when I finish chemotherapy, I have side effects, as you said; when I focus on something, I forget the feeling. I'll feel better then. I think it's also better because when I focus on this, I forget that feeling. Or maybe, like two days ago, I told*

*you I had pain here (pointing to her stomach). Then, when I practised those two days, I focused on listening to music and fell asleep. Normally, I have a hard time sleeping, and I can't sleep. If I can't sleep, I practice like that. Then I try to practice and see if I listen to the music, the music is also gentle, I think it's easier to sleep, I don't think so then I listen to it, and then I finish practising, I go to sleep. Every day, it's very difficult to sleep like I have pain, the day I told my sister that I didn't understand why this time I went to transmit it for some reason, but I kept hurting all the time. The pain here is like I have a stomach ache; I go to sleep so hard I can't sleep. When I'm done practising, I feel like I relax, I listen to the music, and then I fall asleep, I forget the pain (P4)"*

## 4. Discussion

This pilot randomized controlled trial provided a novel way that combined MI with PMR to solve psychological issues commonly faced by breast and gynaecological cancer patients receiving chemotherapy. The pilot study results showed a promisingly high feasibility of the combined intervention and a potentially effective strategy for anxiety, depression and stress management in the target population.

Our study supported the feasibility of the study with high recruitment and consent rate and acceptable attrition rate (16.7%), which was similar to the previous studies with the attrition rate ranging from 4 to 25% [36–38]. However, the reasons for dropout in our study were due to objective factors such as changing hospital or changing chemotherapy schedule because of Covid-19 infection, while a large number of dropouts in the previous studies were due to discontinued intervention [36–38]. The most challenging part of a home-based intervention is how to manage and boost the participants' adherence at home. In our study, apart from the commonly used strategies such as phone calls and distributing guidelines and forms serving as a reminder to participants, we sought support from family members of the participants. Family members were assigned to remind the participants to self-practise the intervention at home. Furthermore, the high adherence rate in our study could also be attributed to the materials and Mp3 recording we have prepared. These materials were carefully assessed by an expert panel, which ensured the scientific standard and adapted cultural elements of the population, such as language, music genre, and tempo.

The current study found that the participants in the intervention group, on average, had greater reduction in anxiety, depression and stress at T1 and T2. The results were consistent with a previous study conducted by Zhou, Li [39], which showed that the combined MI and PMR could reduce anxiety and depression in female breast cancer patients after a radical mastectomy. Although no study evaluated MI combined with PMR on stress among the cancer population, it was assessed in several other populations in the previous studies. For example, MI combined with PMR appeared to be useful in reducing stress in intensive care nurses [40].

The mechanism of music intervention on anxiety, depression and stress is based on the idea that music can influence the body and mind in many ways. Music's rhythm and melody can lower the stress hormone cortisol levels through acting on the hippocampus and peripheral nervous system [41]. A study found a statistically significant decrease in salivary cortisol levels on breast cancer patients after listening to music [41]. Besides, listening to music had been found to activate the reward system in the brain, releasing dopamine and serotonin [42,43], which contribute to reducing anxiety, depression and stress [44]. Music can solve psychological issues by relaxing the mind, delivering distraction from worries and pessimistic thoughts, and allowing the listener to focus on something more positive [45].

PMR is commonly used to manage stress-related issues. When faced with stress, the body produces a series of reactions, including the tense response of muscle groups [46]. The

principle of PMR is based on the tension and relaxation of muscles to change the emotions of the body. Importantly, muscle is supplied by both motor and sensory nerves. Muscle contractions cause electrical impulses and are transmitted to the central nervous system [46]. It causes disturbances in the central nervous system and causes emotional changes. Adding more pressure to the muscles, then suddenly relaxing, makes the emotional disturbances disappear for the moment [46].

The current study found greater improvements in QoL in the intervention group than in the control group at T1 and T2, which was in line with the previous studies [47,48]. The QoL of the patients includes four components physical, social/family, emotional and functional well-being. Evidence indicated that MI and PMR had not only a positive impact on emotional well-being, they but also could improve physical, social/ family and functional well-being. MI could release pain for cancer patients [49]. Previous systematic review of six RCTs revealed that PMR was an effective strategy for preventing and reducing nausea and vomiting caused by chemotherapy in cancer patients [50]. The effects of MI and PMR on reducing insomnia for cancer patients were found in previous studies [51–54]. The results of qualitative analysis in the current study reported the impact of MI combined with PMR on reducing nausea, vomiting, pain, and insomnia.

## Strength and limitation

A well-designed pilot randomized controlled trial was adopted in the current study with rigorous randomization allocation and blinded assessor. The intervention was developed basing the evidence and was evaluated by an expert panel. Multiple strategies were used to enhance the adherence rate of the participants at home.

However, there are several limitations that need to be acknowledged. Firstly, due to the time limitation and COVID-19 impact, we only conducted the pilot with a small sample size. Larger sample size in further studies is warranted to evaluate the effectiveness of the intervention among women with breast and gynaecological cancer receiving chemotherapy. Secondly, this was conducted in a single centre, which limited the generalizability of our study results. In future studies, it is necessary to conduct in multiple centres. Finally, it is impossible to blind participants or interveners because of the nature of the study [55], which may be a potential risk of bias influencing the study results.

## Conclusion

Music intervention combined with progressive muscle relaxation was feasible and acceptable to women with breast and gynaecological cancer undergoing chemotherapy. The combined intervention may help manage anxiety, depression and stress in the target population, however, insufficient power pilot study findings should be interpreted with caution.

## Supporting information

**S1 Checklist. CONSORT 2010 checklist of information to include when reporting a randomised trial\*.**
(DOC)

**S1 File. Tensing instruction of 16 muscle groups.**
(PDF)

**S2 File. Study protocol.**
(DOCX)

## Acknowledgments

The authors would like to acknowledge all staff in Chemotherapy Unit H5, International Cooperation & Scientific Research Unit, Hanoi Oncology hospital for their support during data collection. We would like to thank Dr. Thapa Chura Bahadur for his support in language editing. We would like to express our sincere thanks to Prof. Debra Burns, School of Engineering and Technology, US; Prof. John Mondanaro, The Louis Armstrong Department of Music Therapy, US; Prof. Zehra Gok Metin Hacettepe University Faculty of Nursing, Internal Medicine Nursing Department, Turkey, Ms Hoang Thi Thanh Hue, Vietnam-France Psychology Institute, Vietnam for help with content evaluation of the intervention.

## Author Contributions

**Conceptualization:** Khanh Thi Nguyen, Carmen W. H. Chan.

**Data curation:** Khanh Thi Nguyen, Huong T. X. Hoang, Quang V. Bui.

**Formal analysis:** Kai C. Choi.

**Methodology:** Khanh Thi Nguyen, Huong T. X. Hoang, Dorothy N. S. Chan, Carmen W. H. Chan.

**Supervision:** Carmen W. H. Chan.

**Writing – original draft:** Khanh Thi Nguyen.

**Writing – review & editing:** Khanh Thi Nguyen, Huong T. X. Hoang, Quang V. Bui, Dorothy N. S. Chan, Kai C. Choi, Carmen W. H. Chan.

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
