## [Decision Letter · Decision Letter 0]

19 Dec 2022

PONE-D-22-31423Effects of Music Intervention combined with Progressive Muscle Relaxation on Anxiety, Depression, Stress and Quality of Life among Women with cancer Receiving Chemotherapy: A Pilot Randomized Controlled TrialPLOS ONE

Dear Dr. Chan,

Thank you for submitting your manuscript to PLOS ONE. After careful consideration, we feel that it has merit but does not fully meet PLOS ONE’s publication criteria as it currently stands. Therefore, we invite you to submit a revised version of the manuscript that addresses the points raised during the review process.

We look forward to receiving your revised manuscript.

Kind regards,

Walid Kamal Abdelbasset, Ph.D.

Academic Editor

PLOS ONE

Journal Requirements:

Reviewers' comments:

Reviewer's Responses to Questions

**Comments to the Author**

1. Is the manuscript technically sound, and do the data support the conclusions?

Reviewer #1: Partly

Reviewer #2: Yes

Reviewer #3: Yes

Reviewer #4: Yes

2. Has the statistical analysis been performed appropriately and rigorously? 

Reviewer #1: No

Reviewer #2: Yes

Reviewer #3: Yes

Reviewer #4: Yes

3. Have the authors made all data underlying the findings in their manuscript fully available?

Reviewer #1: Yes

Reviewer #2: Yes

Reviewer #3: Yes

Reviewer #4: Yes

4. Is the manuscript presented in an intelligible fashion and written in standard English?

Reviewer #1: Yes

Reviewer #2: Yes

Reviewer #3: Yes

Reviewer #4: Yes

5. Review Comments to the Author

Reviewer #1: Thank you for giving me this opportunity to review this article. The article is well written, though I have some serious concerns regarding the article.

Keywords: use MeSH keywords

Abstract:

1. Include a brief background of the study.

2. Mention the study duration and study setting.

3. Include the character of the study participants.

4. Include the all the study outcome measures.

5. Mention the statistical tests performed for the study.

6. The results should be presented with 95%CI (upper limit – lower limit) for all the variables.

7. The conclusion should be drawn on the basis of the study reports, not on the assumption.

Manuscript

8. Mention the application procedure, it’s merits and demerits with recent references.

9. Mention the gaps monitored by the researcher.

10. Include the clinical significance of this study over clinicians, patients, and researchers after the study hypothesis.

11. Convenience sampling is not an apt method of sampling for RCT.

12. Mention the randomization and allocation of the subjects in detail.

13. Mention who has included the study participants in the trial?

14. Mention about the informed consent information.

15. Mention the detail of intervention as a separate appendix with figures for study replication.

16. Include the study outcome measures and its reliability and validity.

17. Include the reference study for sample size calculation.

18. The samples included in the study is not sufficient enough to generalize the reports.

19. The statistical tests used for the study was not apt to this study – please consult a biostatistician.

20. Mention the test done for homogenous evaluation.

21. The results should be presented with 95%CI (upper limit – lower limit) for all the variables.

22. There is redundant information in the discussion part and not presented in a logical manner.

23. Add more recent references explaining the mechanism of changing outcome variables through the administration of these intervention.

24. The conclusion should be more concise and self-explanatory and drawn on the basis of study reports.

25. Avoid abbreviations in the conclusion.

26. Add more real-time limitations faced by the researcher and the study.

Reviewer #2: Introduction:

. Explain the rationale of the study. Please delete information unrelated to objective so that the section is short and sweet. For example, the first page of introduction may be deleted. Kindly focus on three elements of introduction.

The discussion section needs to be described scientifically. Kindly frame it along the following lines:

i. Main findings of the present study

ii. Comparison with other studies

iii. Implication and explanation of findings

iv. Strengths and limitations

Reviewer #3: Kindly, the authors should review the attched file and see the comments and the required modifiations for the soundness of the manuscript to be published,and Ihope you can do more studies concerning music therapy.

Reviewer #4: Important note: This review pertains only to ‘statistical aspects’ of the study and so ‘clinical aspects’ [like medical importance, relevance of the study, ‘clinical significance and implication(s)’ of the whole study, etc.] are to be evaluated [should be assessed] separately/independently. Further please note that any ‘statistical review’ is generally done under the assumption that (such) study specific methodological [as well as execution] issues are perfectly taken care of by the investigator(s). This review is not an exception to that and so does not cover clinical aspects {however, seldom comments are made only if those issues are intimately / scientifically related & intermingle with ‘statistical aspects’ of the study}. Agreed that ‘statistical methods’ are used as just tools here, however, they are vital part of methodology [and so should be given due importance]. I look at the manuscript in/with statistical view point, other reviewer(s) look(s) at it with different angle so that in totality the review is very comprehensive. However, there should be efforts from authors side to improve (may be by taking clues from reviewer’s comments). Therefore, please do not limit the revision only (with respect) to comments made here.

COMMENTS: Though the topic is very important, as according ABSTRACT the aim of this study is stated as ‘to investigate the feasibility, acceptability and preliminary effects of music intervention combined with progressive muscle relaxation on anxiety, depression, stress, and quality of life’ is little questionable because there is no question of testing/knowing feasibility as well as acceptability of such intervention [which is well established & known in many types of cancer patients]. Do you expect that feasibility and acceptability will differ in cancer patients receiving chemotherapy? May please refer to quoted references in this manuscript (10-17) itself.

This study being ‘pilot’ in nature, sample size is not a big issue. However, [though many things are ignored (loosely looked at / evaluated)] methodological issues need to be very rigorous followed. As said in lines 370-372: “In our study, the non-significant results might be due to the small sample size decreasing the study's statistical power and confidence level” but remember that “Absence of evidence is not evidence of absence” [Altman DG, Bland JM. BMJ volume 311, 1995, p 485 (Reprinted: Australian Veterinary Journal 1996;74, 311)]. {Even when P-value is not significantly lower that is null hypothesis of no difference / no association is not rejected, (in short, result is not significant), that does not amount to evidence of absence i.e., it does not imply that there no difference / no association. It only implies that there is no (i.e., these samples do not provide) [say enough] evidence to prove (rather indicate with certain specified confidence level) the difference / association}. Therefore, conclusion(s) from any study [in which result(s) is/are not significant], should be drawn in the light of this fact.

I am not surprised to read lines like 285-287: “A trend toward a greater reduction was noted in anxiety in the intervention group at T2 with a medium effect size, yet no significant difference was observed between the two groups (g= 0.57, p=0.28)” in backdrop of the fact [other point to mind] that: a pilot trial is not to assess effectiveness (or efficacy) and therefore, no formal hypothesis testing for effectiveness (or efficacy). I do not understand/know, why you say “with a medium effect size” because in studies on such topic generally the ‘effect size’ is very small and so you are unlikely to find any difference while working with such small sample.

Supporting information [file S1] includes CONSORT Checklist, but there you are supposed to give exact page number(s) for each item (last column: Reported on page No.). Due to absence (lack) of this information, it is difficult to check the presence/mention of many important items (though most of important items are covered / described adequately). ‘Fig 1: The Consolidated Standards of Reporting Trial (CONSORT) flow diagram’ is perfectly alright.

According to lines 228-230, “Mann–Whitney U test was used to compare the mean difference of DASS and FACT-G between groups at baseline (T0) and T1, T2, respectively”. Note that since there are three time points which are not independent and so more appropriate test is Wilcoxon’s Signed Rank test. In fact, first apply ‘Kruskal-Wallis’ test [which is non-parametric one-way ANOVA] or preferably ‘Friedman’s test [which is non-parametric two-way ANOVA]. Also consider ‘multiple comparisons’ issue (adjust P-values yielded by Wilcoxon’s Signed Rank test). But because there are two independent groups (experimental Vs control) to be compared ultimately, best is to work on ‘change scores’. In short, for within-group two samples comparison (i.e., Paired Case) use Wilcoxon’s Signed-Ranked test, for comparison of Several Means within groups, use Two-way lay out: Friedman’s test, for between-group two samples comparison (i.e., Un-paired Case) use Mann–Whitney U test, and for comparison of Several Means - One way layout: Kruskal-Wallis’s test.

Further, since Hedges’ g (like Cohen’s d) is biased upwards for small samples (under 50), hope the correction suggested for small sample size is applied. In the context of ‘Table 2: Baseline demographic information of the participants by group assignment’, please read the following para pasted from one famous standard textbook on ‘Medical Research Methodology’.

To provide a description of baseline characteristics is entirely reasonable (since it is clearly important in assessing to whom the results of the trial can be applied), however, statistical comparison of baseline characteristics when random allocation/assignment is used/done [often for good/standard/leading journals these days] is not required, because even if P-value(s) turn(s) out to be significant (while comparing baseline characteristics despite random allocation), it is, by definition, a false positive as you then are supposed to be testing ‘randomization’ then, which in any single trial may not balance all baseline characteristics (particularly when sample sizes are small). Remember that ‘randomization’ is a sort of ‘insurance’ and not a guarantee scheme. Authors may please refer to following articles:

References:

1. Stuart J. Pocock, et al., ‘Subgroup analysis, covariate adjustment and baseline comparisons in clinical trial reporting: current practice and problems’, Statistics in medicine, 2002; 21:2917–2930 [Particularly page 2927]

2. Harrington D, et al., ‘New guidelines for statistical reporting in the journal’, N Engl J Med 2019;381:285-6

[Important message (indirectly/ultimately indicated) from these articles: Never do any comparison with respect to ‘baseline’ characteristics {by applying statistical significance test(s)}, when allocation is done randomly].

However, Statistical comparison [only with respect to important/indicated variables] of baseline characteristics may be performed, to find out if analysis adjustment (say stratified analyses or else) is required with respect to these variables.

As you know, ‘the Fisher’s exact test’ [which is commonly available in many software] is applicable to 2x2 tables only [for larger tables Exact Chi-square test is available – reference ‘A network algorithm for performing Fisher's exact test in r× c contingency tables’ Cyrus R Mehta, Nitin R Patel, Journal of the American Statistical Association, 1983, Volume: 78, Issue: 382, Pages 427-434 ], however, which one is used here (table-2 or even table-3) is not clear as otherwise how you applied when there are more than two categories {examples: ‘education’, ‘Religion’, ‘Income’}? If you have taken some category as reference (which must), what is criteria used to select the reference category? Mind you that this is a scientific/academic document and so all details should be clearly/correctly communicated (do not take readers’ for granted).

Except these few points, the study is alright. However, as pointed out in ‘important note’ above “This review pertains only to ‘statistical aspects’ of the study and so ‘clinical aspects’ should be assessed separately/independently [one should carefully consider/look at the clinical implications of the study as the implications of any study should be valuable, useful, and convincing to scientists of other discipline]. In my opinion, to rescue this article (which is quite possible), some amount of re-vision (re-drafting) may be needed. The respected ‘Editor’ may consider accepting if found ‘clinical implications’ (of this study) valuable. ‘Minor revision’ is recommended.

6. PLOS authors have the option to publish the peer review history of their article (what does this mean?). If published, this will include your full peer review and any attached files.

Reviewer #1: No

Reviewer #2: No

Reviewer #3: No

Reviewer #4: No

---

## [Author Response · Author response to Decision Letter 0]

22 Apr 2023

Thank you very much for the reviewers' valuable comments on our manuscript, which we have now revised accordingly. Below are our responses to the reviewers' comments:

*Please note that all mentioned lines and pages below are on the revised manuscript with the track changes file.

Reviewer 1

Keywords: use MeSH keywords

Response: Thank you so much for your suggestion. We revised the keywords using MESH keywords in lines 82-83, page 3. 

Abstract:

1. Include a brief background of the study.

Response: Thank you so much for your suggestion. A brief background was included in lines 47-49 page 2. 

2. Mention the study duration and study setting.

Response: Thank you so much for your suggestion. The study duration and study setting were included in lines 53-54, page 2. 

3. Include the character of the study participants.

Response: Thank you so much for your suggestion. The character of the study participants was included in lines 54-55, page 2. 

4. Include all the study outcome measures.

Response: Thank you so much for your suggestion. The study outcome measures were included in lines 59-62, page 2.

5. Mention the statistical tests performed for the study.

Response: Statistics were mentioned in lines 64-65, page 2.

6. The results should be presented with 95%CI (upper limit – lower limit) for all the variables.

Response: The effect sizes have now been presented with 95% CIs.

7. The conclusion should be drawn on the basis of the study reports, not on the assumption.

Response: The conclusion was revised in lines 77-80, page 3. 

Manuscript

8. Mention the application procedure, it's merits and demerits with recent references.

Response: Thank you so much for your suggestion. The application procedure, its merits and demerits with recent references were mentioned in lines 138-155, pages 5-6. 

9. Mention the gaps monitored by the researcher.

Response: Thank you so much for your suggestion. The gaps were mentioned in lines 138-155, pages 5-6. 

10. Include the clinical significance of this study over clinicians, patients, and researchers after the study hypothesis.

Response: Thank you so much for your suggestion. The clinical significance of this study was added in lines 164-166, page 6. 

11. Convenience sampling is not an apt method of sampling for RCT.

Response: Since it is infeasible to compile a sampling frame for the target population, we believe convenience sampling would be an appropriate and practical approach.

12. Mention the randomization and allocation of the subjects in detail.

Response: Thank you so much for your suggestion. The randomization and allocation were included in lines 189-196, page 7.

13. Mention who has included the study participants in the trial?

Response: Thank you so much for your suggestion. The person who included the participants was mentioned in lines 197-198, page 7.

14. Mention about the informed consent information.

Response: Thank you so much for your comments. The informed obtained consent form from participants was mentioned in line 197, page 7. 

15. Mention the detail of the intervention as a separate appendix with figures for study replication.

Response: Thank you so much for your suggestion. A detailed tensing instruction of 16 muscle groups in the sequence was provided in the supporting information (S2). 

16. Include the study outcome measures and its reliability and validity.

Response: Thank you so much for your comments. The study outcome measures and its reliability and validity were described in lines 247-263, pages 9-10. 

17. Include the reference study for sample size calculation.

Response: Thank you so much for your suggestion. The reference study for sample size calculation was cited and presented in the reference as "Julious SA. Sample size of 12 per group rule of thumb for a pilot study. Pharm Stat. 2005;4(4):287-91."

18. The samples included in the study is not sufficient enough to generalize the reports.

Response: We agree that the findings of our small sample pilot study may not be generalizable. We just want to provide some findings regarding the feasibility and acceptability as well as the preliminary effects of the music combined with progressive muscle relaxation intervention. We have reminded the readers to interpret our findings with cautions in our conclusion.

19. The statistical tests used for the study was not apt to this study – please consult a biostatistician.

Response: Thank you so much for your suggestion. We invited a statistician to be a co-author and review the manuscript and help address the statistical issues.

20. Mention the test done for homogenous evaluation.

Response: In view of the small sample size of the present pilot study and hence a likely underpowered study, we have now decided not to do any statistical test. 

21. The results should be presented with 95%CI (upper limit – lower limit) for all the variables.

Response: Please see our response to comment #6 above.

22. There is redundant information in the discussion part and not presented in a logical manner.

Response: Thank you so much for your suggestion. The discussion part has been revised, some parts were rejected. 

23. Add more recent references explaining the mechanism of changing outcome variables through the administration of these intervention.

Response: Thank you so much for your suggestion. The mechanism of changing outcomes was discussed in lines 450-466, page 20. 

24. The conclusion should be more concise and self-explanatory and drawn on the basis of study reports.

Response: Thank you so much for your suggestion. The conclusion was revised in lines 492-498, pages 21-22. 

25. Avoid abbreviations in the conclusion.

Response: Thank you so much for your suggestion. No abbreviation was present in conclusion. 

26. Add more real-time limitations faced by the researcher and the study.

Response: Thank you so much for your suggestion. The real-time limitations was added in the limitation section in lines 483-485, page 21. 

Reviewer 2

1.Introduction: Explain the rationale of the study. Please delete information unrelated to objective so that the section is short and sweet. For example, the first page of introduction may be deleted. Kindly focus on three elements of introduction.

Response: Thank you so much for your suggestion. Some unnecessary parts of the introduction were rejected in lines 100-110, 118-122, pages 3-4. And the introduction was revised following the guidance. 

2. The discussion section needs to be described scientifically. Kindly frame it along the following lines:

i. Main findings of the present study

ii. Comparison with other studies

iii. Implication and explanation of findings

iv. Strengths and limitations

Response: Thank you so much for your suggestion. The discussion was revised following the above guidance in lines 396-490, pages 18-21. 

Reviewer 3

Abstract:

1. The background needs to be added

Response: Thank you so much for your suggestion. A brief background was included in lines 47-48 page 2.

2.Demographic profile of patients is not mentioned.

Response: Thank you so much for your suggestion. The demographic profile of patients were included in lines 54-55, page 2.

Main text:

3.Participants: Why you choose this age limit? And did you find cases at this limit? I think breast cancer in this age will be rare.

Response: Thank you for your questions. This pilot study focused on adult breast and gynaecological cancer patients. It is very rare to see younger age people with BGC cancer. However, it is reported cases in adolescents, even children. 

For example:

Gutierrez, J. C., Housri, N., Koniaris, L. G., Fischer, A. C., & Sola, J. E. (2008, 2008/06/15/). Malignant Breast Cancer in Children: A Review of 75 Patients. Journal of Surgical Research, 147(2), 182-188. https://doi.org/https://doi.org/10.1016/j.jss.2008.03.026

McDivitt RW, Stewart FW. Breast Carcinoma in Children. JAMA. 1966;195(5):388–390. doi:10.1001/jama.1966.03100050096033

4.Study sample: Explain why this type of sampling methods you used?

Response: Since it is infeasible to compile a sampling frame for the target population, we believe convenience sampling would be an appropriate and practical approach.

5.Study sample: Why this method you can use more accurate method for calculation to void biase

Response: The preliminary effects of the intervention on the outcomes have now been estimated by bias-corrected Hedges'g effect sizes.

Reviewer 4:

1.Though the topic is very important, as according ABSTRACT the aim of this study is stated as 'to investigate the feasibility, acceptability and preliminary effects of music intervention combined with progressive muscle relaxation on anxiety, depression, stress, and quality of life' is little questionable because there is no question of testing/knowing feasibility as well as acceptability of such intervention [which is well established & known in many types of cancer patients]. Do you expect that feasibility and acceptability will differ in cancer patients receiving chemotherapy? May please refer to quoted references in this manuscript (10-17) itself.

Response: Yes, although the feasibility and acceptability of music or progressive muscle relaxation interventions have been well established in various types of cancer patients, the combination of both types of intervention has not been well documented, particularly in breast and gynaecological cancer patients undergoing chemotherapy. 

2. This study being 'pilot' in nature, sample size is not a big issue. However, [though many things are ignored (loosely looked at / evaluated)] methodological issues need to be very rigorous followed. As said in lines 370-372: "In our study, the non-significant results might be due to the small sample size decreasing the study's statistical power and confidence level" but remember that "Absence of evidence is not evidence of absence" [Altman DG, Bland JM. BMJ volume 311, 1995, p 485 (Reprinted: Australian Veterinary Journal 1996;74, 311)]. {Even when P-value is not significantly lower that is null hypothesis of no difference / no association is not rejected, (in short, result is not significant), that does not amount to evidence of absence i.e., it does not imply that there no difference / no association. It only implies that there is no (i.e., these samples do not provide) [say enough] evidence to prove (rather indicate with certain specified confidence level) the difference / association}. Therefore, conclusion(s) from any study [in which result(s) is/are not significant], should be drawn in the light of this fact.

Response: Thank you for the comments. As the sample size of our pilot was small and likely underpowered to detect difference of clinical relevance. We have now decided not to do any statistical test and simply use bias corrected Hedges' g effect sizes to preliminarily estimate the effects of the intervention on the outcomes.

3. I am not surprised to read lines like 285-287: "A trend toward a greater reduction was noted in anxiety in the intervention group at T2 with a medium effect size, yet no significant difference was observed between the two groups (g= 0.57, p=0.28)" in backdrop of the fact [other point to mind] that: a pilot trial is not to assess effectiveness (or efficacy) and therefore, no formal hypothesis testing for effectiveness (or efficacy). I do not understand/know, why you say "with a medium effect size" because in studies on such topic generally the 'effect size' is very small and so you are unlikely to find any difference while working with such small sample.

Response: Thank you for the advice and comments. Given the small sample size and hence low precision of effect estimation, we agree that the previous interpretations could be somewhat overstated. We have now only provided the effect size estimates and their 95% confidence intervals and avoided making such kinds of interpretations. 

4. Supporting information [file S1] includes CONSORT Checklist, but there you are supposed to give exact page number(s) for each item (last column: Reported on page No.). Due to absence (lack) of this information, it is difficult to check the presence/mention of many important items (though most of important items are covered / described adequately). 'Fig 1: The Consolidated Standards of Reporting Trial (CONSORT) flow diagram' is perfectly alright.

Response: We have now added the exact page numbers (fit with the manuscript file without track changes) for the CONSORT items.

5.According to lines 228-230, "Mann–Whitney U test was used to compare the mean difference of DASS and FACT-G between groups at baseline (T0) and T1, T2, respectively". Note that since there are three time points which are not independent and so more appropriate test is Wilcoxon's Signed Rank test. In fact, first apply 'Kruskal-Wallis’ test [which is non-parametric one-way ANOVA] or preferably ‘Friedman’s test [which is non-parametric two-way ANOVA]. Also consider ‘multiple comparisons’ issue (adjust P-values yielded by Wilcoxon’s Signed Rank test). But because there are two independent groups (experimental Vs control) to be compared ultimately, best is to work on ‘change scores’. In short, for within-group two samples comparison (i.e., Paired Case) use Wilcoxon’s Signed-Ranked test, for comparison of Several Means within groups, use Two-way lay out: Friedman’s test, for between-group two samples comparison (i.e., Un-paired Case) use Mann–Whitney U test, and for comparison of Several Means - One way layout: Kruskal-Wallis’s test.

Response: Thank you for the suggestions. Please refer to our response to comment #2 above.

6. Further, since Hedges’ g (like Cohen’s d) is biased upwards for small samples (under 50), hope the correction suggested for small sample size is applied. In the context of ‘Table 2: Baseline demographic information of the participants by group assignment’, please read the following para pasted from one famous standard textbook on ‘Medical Research Methodology’.

To provide a description of baseline characteristics is entirely reasonable (since it is clearly important in assessing to whom the results of the trial can be applied), however, statistical comparison of baseline characteristics when random allocation/assignment is used/done [often for good/standard/leading journals these days] is not required, because even if P-value(s) turn(s) out to be significant (while comparing baseline characteristics despite random allocation), it is, by definition, a false positive as you then are supposed to be testing ‘randomization’ then, which in any single trial may not balance all baseline characteristics (particularly when sample sizes are small). Remember that ‘randomization’ is a sort of ‘insurance’ and not a guarantee scheme. Authors may please refer to following articles:

References:

1. Stuart J. Pocock, et al., ‘Subgroup analysis, covariate adjustment and baseline comparisons in clinical trial reporting: current practice and problems’, Statistics in medicine, 2002; 21:2917–2930 [Particularly page 2927]

2. Harrington D, et al., ‘New guidelines for statistical reporting in the journal’, N Engl J Med 2019;381:285-6

[Important message (indirectly/ultimately indicated) from these articles: Never do any comparison with respect to ‘baseline’ characteristics {by applying statistical significance test(s)}, when allocation is done randomly].

However, Statistical comparison [only with respect to important/indicated variables] of baseline characteristics may be performed, to find out if analysis adjustment (say stratified analyses or else) is required with respect to these variables.

Response: Thank you for the comments and references. We agree not to do statistical comparisons between groups regarding baseline characteristics, particularly without prior knowledge about potentially important prognostic or confounding factors. 

7.As you know, ‘the Fisher’s exact test’ [which is commonly available in many software] is applicable to 2x2 tables only [for larger tables Exact Chi-square test is available – reference ‘A network algorithm for performing Fisher's exact test in r× c contingency tables’ Cyrus R Mehta, Nitin R Patel, Journal of the American Statistical Association, 1983, Volume: 78, Issue: 382, Pages 427-434 ], however, which one is used here (table-2 or even table-3) is not clear as otherwise how you applied when there are more than two categories {examples: ‘education’, ‘Religion’, ‘Income’}? If you have taken some category as reference (which must), what is criteria used to select the reference category? Mind you that this is a scientific/academic document and so all details should be clearly/correctly communicated (do not take readers’ for granted).

Response: Thank you for the advice. We will bear in mind the suggestions for our future reports, as in view of the small sample size of the present pilot study and hence a likely underpowered study, we have now decided not to do any statistical tests on baseline comparison.

---

## [Decision Letter · Decision Letter 1]

13 Sep 2023

PONE-D-22-31423R1Effects of Music Intervention combined with Progressive Muscle Relaxation on Anxiety, Depression, Stress and Quality of Life among Women with cancer Receiving Chemotherapy: A Pilot Randomized Controlled TrialPLOS ONE

Dear Dr. Chan,

Thank you for submitting your manuscript to PLOS ONE. After careful consideration, we feel that it has merit but does not fully meet PLOS ONE’s publication criteria as it currently stands. Therefore, we invite you to submit a revised version of the manuscript that addresses the points raised during the review process.

Your revised manuscript has been reviewed and the reviewers have confirmed that previous comments have been addressed. Please see review comments below. Although Reviewer #1 has asked that the study type be changed to a pilot study, this is not a requirement.

However, we note that you have not uploaded a copy of detailed trial study protocol. This is a requirement for clinical trials to be published, including as a Study Protocol article type. Please see the section "Journal Requirements" for further details.

We look forward to receiving your revised manuscript.

Kind regards,

Marianne Clemence

Staff Editor

PLOS ONE

Journal Requirements:

Please upload a copy of your trial study protocol as a supporting information file. By the study protocol, we mean the complete and detailed plan for the conduct and analysis of the trial that the ethics committee approved before the trial began. Please send this in the original language. If this is in a language other than English, please also provide a translation. Please detail any deviations from this study protocol in the Methods section of your manuscript. Your study protocol will be made available to the editors and reviewers, and will be published as supporting information with your manuscript if accepted for publication. (If you do not agree to this, we will not be able to publish your manuscript).

Reviewers' comments:

Reviewer's Responses to Questions

**Comments to the Author**

1. If the authors have adequately addressed your comments raised in a previous round of review and you feel that this manuscript is now acceptable for publication, you may indicate that here to bypass the “Comments to the Author” section, enter your conflict of interest statement in the “Confidential to Editor” section, and submit your "Accept" recommendation.

Reviewer #1: All comments have been addressed

Reviewer #2: All comments have been addressed

Reviewer #4: All comments have been addressed

2. Is the manuscript technically sound, and do the data support the conclusions?

Reviewer #1: Yes

Reviewer #2: Yes

Reviewer #4: (No Response)

3. Has the statistical analysis been performed appropriately and rigorously? 

Reviewer #1: Yes

Reviewer #2: Yes

Reviewer #4: (No Response)

4. Have the authors made all data underlying the findings in their manuscript fully available?

Reviewer #1: Yes

Reviewer #2: Yes

Reviewer #4: (No Response)

5. Is the manuscript presented in an intelligible fashion and written in standard English?

Reviewer #1: Yes

Reviewer #2: Yes

Reviewer #4: (No Response)

6. Review Comments to the Author

Reviewer #1: Dear authors,

I really appreciate all your efforts to address my comments in a very positive manner.

I have a suggestion - please change the study type to - A pilot study. It is not a RCT.

Regards

Reviewer #2: thanks for your replay

Reviewer #4: COMMENTS: All the comments are answered and positively attended. I recommend the acceptance because the manuscript has now achieved the acceptable level in my opinion.

7. PLOS authors have the option to publish the peer review history of their article (what does this mean?). If published, this will include your full peer review and any attached files.

Reviewer #1: **Yes: **Gopal Nambi

Reviewer #2: **Yes: **Ahmed Abdelmoniem Ibrahim

Reviewer #4: **Yes: **Dr. Sanjeev Sarmukaddam

---

## [Author Response · Author response to Decision Letter 1]

29 Sep 2023

Thank you very much for the valuable comments on our manuscript, which we have now revised accordingly. Below are our responses to comments of the editors and reviewers:

A. Journal Requirements:

Response: Thank you for your reminder. The reference list was carefully checked that it includes published date and a database's unique identifiers (PMID, PMCID, or DOI). There are no retracted papers was cited in the reference list. 

2. Please upload a copy of your trial study protocol as a supporting information file. By the study protocol, we mean the complete and detailed plan for the conduct and analysis of the trial that the ethics committee approved before the trial began. Please send this in the original language. If this is in a language other than English, please also provide a translation. Please detail any deviations from this study protocol in the Methods section of your manuscript. Your study protocol will be made available to the editors and reviewers, and will be published as supporting information with your manuscript if accepted for publication. (If you do not agree to this, we will not be able to publish your manuscript).

Response: The study protocol has been uploaded as supporting information file labeled S3. 

B. Reviewer’s comments: 

1. Reviewer 1: I have a suggestion - please change the study type to - A pilot study. It is not a RCT.

Thank you so much for your suggestion. Although this is a pilot study, it follows a randomized controlled trial approach. We believe that the original title "A Pilot Randomized Controlled Trial" more accurately reflects the study design and provides clarity. Thus, we have decided to maintain the original title.

---

## [Editor Report · Decision Letter 2]

5 Oct 2023

Effects of Music Intervention combined with Progressive Muscle Relaxation on Anxiety, Depression, Stress and Quality of Life among Women with cancer Receiving Chemotherapy: A Pilot Randomized Controlled Trial

PONE-D-22-31423R2

Dear Dr. Chan,

We’re pleased to inform you that your manuscript has been judged scientifically suitable for publication and will be formally accepted for publication once it meets all outstanding technical requirements.

Kind regards,

Marianne Clemence

Staff Editor

PLOS ONE
---

## [Editor Report · Acceptance letter]

26 Oct 2023

PONE-D-22-31423R2 

Effects of Music Intervention combined with Progressive Muscle Relaxation on Anxiety, Depression, Stress and Quality of Life among Women with cancer Receiving Chemotherapy: A Pilot Randomized Controlled Trial 

Dear Dr. Chan:

I'm pleased to inform you that your manuscript has been deemed suitable for publication in PLOS ONE. Congratulations! Your manuscript is now with our production department. 

Kind regards, 

on behalf of

Dr Marianne Clemence 

Staff Editor

PLOS ONE